# Multipurpose Bio-Monitored Integrated Circuit in a Contact Lens Eye-Tracker

**DOI:** 10.3390/s22020595

**Published:** 2022-01-13

**Authors:** Loïc Massin, Cyril Lahuec, Fabrice Seguin, Vincent Nourrit, Jean-Louis de Bougrenet de la Tocnaye

**Affiliations:** 1Optics Department, Institut Mines-Télécom Atlantique, Technopôle Brest Iroise, CS 83818, CEDEX 03, 29238 Brest, Brittany, France; loic.massin@imt-atlantique.fr (L.M.); fabrice.seguin@imt-atlantique.fr (F.S.); vincent.nourrit@imt-atlantique.fr (V.N.); jl.debougrenet@imt-atlantique.fr (J.-L.d.B.d.l.T.); 2Laboratoire des Sciences et Techniques de l’Information, de la Communication et de la Connaissance, UMR 6285, 29238 Brest, Brittany, France

**Keywords:** instrumented contact lens, eye-tracker, blink detection, blink interpretation, CMOS, ASIC

## Abstract

We present the design, fabrication, and test of a multipurpose integrated circuit (Application Specific Integrated Circuit) in AMS 0.35 µm Complementary Metal Oxide Semiconductor technology. This circuit is embedded in a scleral contact lens, combined with photodiodes enabling the gaze direction detection when illuminated and wirelessly powered by an eyewear. The gaze direction is determined by means of a centroid computation from the measured photocurrents. The ASIC is used simultaneously to detect specific eye blinking sequences to validate target designations, for instance. Experimental measurements and validation are performed on a scleral contact lens prototype integrating four infrared photodiodes, mounted on a mock-up eyeball, and combined with an artificial eyelid. The eye-tracker has an accuracy of 0.2°, i.e., 2.5 times better than current mobile video-based eye-trackers, and is robust with respect to process variations, operating time, and supply voltage. Variations of the computed gaze direction transmitted to the eyewear, when the eyelid moves, are detected and can be interpreted as commands based on blink duration or using blinks alternation on both eyes.

## 1. Introduction

Recent advances in wearable electronics, combined with wireless communications, have triggered a huge interest in the development of smart contact lenses [1]. Currently, they are capable of monitoring physiological information to aid with the treatment of pathologies (e.g., intraocular pressure for glaucoma or tear film glucose concentration for diabetics) and may be the next step in augmented reality (cf., retinal display concepts such as the Mojo lens). Hence, contact lenses should not be considered only as single task device but as an integrating platform embedding several functions simultaneously. Our paper examines the use of an ASIC to implement multipurpose tasks on the same contact lens, i.e., here eye-tracking and blink detection.

In parallel, eye-tracking has become a common tool in many fields, from cognitive science to human-computer interactions passing by medical diagnosis [2], driver assistance [3], and head-mounted displays with augmented/virtual reality [4]. Video-based eye-trackers are currently the most widely used approach. One or multiple cameras acquire images of the eyes, usually illuminated in infrared, and the gaze direction is estimated by image processing using various methods [5]. The video-based approach owns its success to its noninvasive nature and constant progresses in terms of imaging, computing power, and image processing. However, various imaging conditions (camera resolution, iris color, lighting conditions, etc.) can significantly decrease performance. Moreover, the necessary powerful computers and fast camera make them difficult to integrate in demanding environments. Accurate gaze tracking typically requires head immobilization, the use of several cameras and IR sources, or lengthy calibration [4]. Thus, over the last decade, the reported accuracy of head mounted eye-trackers remains above 0.5° [2,3,4,5,6,7].

An approach to improve gaze tracking accuracy is the integration of functions in scleral contact lens (SCL) [8,9,10]. In this system, photodetectors (PTDs) encapsulated into an SCL, are illuminated by an IR source placed on an eyewear. The gaze direction is computed using the PTDs response by an integrated circuit (ASIC) implemented into the lens and powered by an inductive link. The result is then wirelessly sent to the eyewear by near-field communication (NFC), also used to power the ASIC. The ASIC is activated as soon as the inductive link is established and transmits the digitized gaze direction to the eyewear, Figure 1a. Hence, no action from the user is required.

Such a device could also be used to detect blinks. Blinks may provide important information on cognitive processing but it could also serve as the basis for a specific human-machine interface (HMI) [11]. For instance, in demanding environments (e.g., surgery, driving) the user could blink to select a function among several implemented into the SCL-eyewear pair (e.g., eye-tracking, displaying information, etc.), while keeping his hands free for other tasks [12]. Such a system should obviously be able to differentiate between involuntary and deliberate blinks. A blinking time period is usually 0.1 s to 0.2 s, and therefore one can consider that, when the blinking time period exceeds 0.5 s, the blinking is a conscious one.

This paper extends works presented in [9] by adding this HMI to the system. The HMI is implemented using the ASIC designed to continuously compute the gaze direction. The blinking command is detected on the eyewear from the data sent by the SCL by a simple algorithm. By analyzing blink durations or blink sequences on both eyes, a voluntary command can be detected. This command could be used to validate whether the gaze direction data are recorded or not or to validate a target designation. The number of possible combinations is increased when considering both eyes to design more complex HMIs. Moreover, we present performance measurements (accuracy, precision, robustness, energy harvesting, among others) of the full system (ASIC encapsulated into a scleral lens and eyewear) in realistic test set-ups whereas [9] presented only simulation results of the ASIC.

## 2. Materials and Methods

The system is composed of the instrumented 16.5 mm diameter SCL and the eyewear. The instrumented SCL comprises a gaze-computing ASIC, four infrared Silonex SLCD-61N8 photodetectors (a sensitivity of *R*λ = 0.55 A/W at λ = 940 nm), and a NXP NFC NTAG NT3H110. Based on the PTDs response, a 1.8 V subthreshold biased 0.35 µm CMOS ASIC computes the centroid of the IR received. As the eye moves, the centroid changes accordingly [9]. Two identical current-mode subthreshold analog CMOS cores perform the centroid computations, one per direction, horizontal (*θ*) and vertical (*φ*), only activated when required to save power. The results are then sequentially digitized by a voltage-mode 12-bit successive-approximation register analog-to-digital converter (SAR ADC). The operating frequency is 12.5 kHz to meet the data rate requirement of at least 500 Hz, and is provided by the output signal divided by 32 of a 400 kHz Wien bridge harmonic oscillator. Since the centroid is output as a current and the SAR ADC is voltage mode, a 1 MΩ resistor R_I–V_ performs the current-to-voltage conversion, having a linear response over a wide dynamic range. In addition, a subthreshold folded-cascode amplifier implements a unity gain buffer for impedance matching. After digitization, both coordinates are sequentially transmitted via the NFC tag, Figure 1b. The time diagram of the ASIC is shown in Figure 2.

Figure 3 shows a microphotograph of the fabricated ASIC. Supplied by a 1.8 V voltage source, the ASIC consumes an average of 137 µW (versus 177 µW, if the centroid computations blocks were on all the time).

The centroid computation yields an inherent error since the eye is a sphere and not a plane. This predictive error is easily corrected by a lookup table (LUT) prefilled with 1024 values uniformly distributed between ±20° by means of a theoretical study, allowing each calculated angle to correspond to a real angle. For system robustness purposes, the LUT contains more corrected values than the typical ±16° angle range for human eye movements. Moreover, to cater for other sources of impairment (fabrication process variations, eyewear–SCL misalignments, etc.) a simple but robust, efficient calibration procedure is implemented [9]. Monte Carlo (MC) simulations showed that after calibration and LUT compensation, system accuracy is 0.2° [9].

The eyewear is equipped with an NFC reader and microcomputing unit implementing, for power consumption considerations, both the LUT and the calibration, the blink detection and interpretation algorithm which triggers commands, Figure 1b. The eyewear electronics is emulated by a Raspberry Pi 3 B+ card running Python scripts for calibration, tabulated correction of the eye-tracker function [9] and the blink detection and interpretation algorithm. Figure 4 shows the prototype electronics before encapsulation in an SCL (a) and after (b) on a mock-up eye. The substrate is a 200 µm thick filled polyimide with copper tracks of thickness 35 µm. To simplify the fabrication, the NFC TAG chip is not encapsulated but is wire-connected to provide the ASIC its power and collect the gaze direction data (hence the flexible ribbon). Figure 5 shows the NFC part of the test bench, the other end of the flexible ribbon is clearly seen. The primary coil on the eyewear is excited with a 13.56 MHz 140 mW source. The lens-size secondary coil is placed at 13 mm from the primary one, which corresponds to the typical eye-to-spectacles distance.

The instrumented SCL is illuminated by an infrared (IR) Lambertian light source using a single IR LED (Vishay TSAL6100, Vishay Intertechnology, Inc., Malvern, PA, USA) at λ = 940 nm [9] associated with a beam splitter to deport the light source on the eyewear so as it is not in the field of view, Figure 6.

Blinking is primarily accomplished by the upper lid, the lower one remains essentially idle. The upper lid displacement on the cornea is about 7 mm with different speeds during opening and closing action, 70 and 160 mm/s in average, respectively [13]. The eyelid mainly blocks visible light, with less than 5% transmittance for wavelengths below 600 nm. Moving toward the near infrared, the transmittance of the eyelid increases to around 80% for λ > 900 nm [14].

An artificial upper lid with optical characteristics similar to the human one was designed and assembled. Figure 7 illustrates the set-up, (a) side view and (b) front view. A 25 × 25 mm^2^ color density filter with a 76% transmittance in the near IR (940 nm) is used as material for the eyelid. The lid movements are emulated using a servomotor (Analog RE3LY S-0008, Conrad Electronic, Hirschau, Germany) to lower and lift the color density in front of the mock-up eye. The color density is placed on 30 mm radius disc centered on the servorotation axis. The number of rotations per min (rpm) of the servomotor is set so as to correspond to the upper lid displacements speeds, e.g., 36 and 16 rpm for closing and opening, respectively, yielding artificial eyelid displacement speeds of 160 and 70 mm/s, respectively. The servomotor movements are controlled by an Arduino Uno card, running basic programs defining the blinking parameters, e.g., number, duration, frequency, etc. Figure 7c,d shows the actual realization of the artificial upper lid. To facilitate assembly, the IR light source is placed in front of the mock-up eye, the eyewear with the deported IR source preventing the correct motion of the artificial lid. For experiments, the light source is in front of the lens to ensure that it has the right source type and placed at *d*0 = 13 mm from the prototype. Using an optical power meter (Thorlabs PM100D), the irradiance at the surface of the lens is 16.4 µW per mm^2^, in accordance with eye safety regulation [6].

The eyewear is implemented as a Raspberry Pi 3 B+ card. The gaze direction, directly read from the I2C ports of the tag to ease prototyping (see Figure 5), is first calibrated and corrected according to the algorithm described in [9] before being processed to detect the blinking command. When the eyelid closes, it covers gradually the four PTDs, starting from the top. This rapidly modifies the value of the computed centroid. When the eyelid covers all the PTDs, the centroid value returns to its initial value (since the influence of the eyelid then disappears in the centroid calculation). There is thus a brief but significant variation, a glitch, in the time sequence of the received centroid values at the beginning of the blink. The behavior is similar when the eyelid opens. To detect these variations a *z*-score algorithm is implemented on the Raspberry Pi. Basically, it computes the distance of a measured gaze direction to the mean value of the gaze direction computed over a sliding time window, Equation (1), considering only the current horizontal eye angle *θ*.
(1)$$Z=\frac{\theta -\mu }{\sigma },$$
where *μ* is the average value of the previous 500 computed values of *θ* and *σ* the standard deviation of these 500 values of *θ*. Then, the absolute value of *Z* is compared to a threshold, chosen from several measurements, which is 4.4; if |*Z*| > 4.4, then the glitch is detected.

Once the glitches are detected, the program can discriminate between voluntary and reflex blinks based on their durations.

## 3. Results

### 3.1. Eye-Tracker Performance

The eye-tracker is first tested using the set-up shown in Figure 6b, with the deported IR light source and the NFC tag providing 1.8 V voltage supply to the ASIC. The test bench is set in room with no natural light but with the mock-up eye rotating within the typical angle range for human eye movements, i.e., from −16° to +16° by steps of 2°. The digitized gaze direction, represented by horizontal angle (*θ*) and vertical angle (*φ*), is transmitted to the eyewear then applying in real time the LUT correction. The resulting gaze direction accuracy only taking the *θ* angle into account is given in Figure 8 (a similar result is obtained for *φ*). This figure shows the accuracy obtained from five different ASICs is consistent with MC analysis represented by the hatched area (250 runs). Repeating 100 times, the accuracy measurement for each point yields a precision of 0.04°.

As most video-based eye-tracker require frequent recalibration, it is interesting to assess the validity duration of the proposed cameraless eye-tracker. The calibration is done at t_0_, and the accuracy is measured then at t_0_, 30 min later and one hour later. The results show the calibration is robust in time (the accuracy remains at most at 0.2°, Figure 9).

### 3.2. Detecting the Eye Blink with the Artificial EyelidASIC Performance

To illustrate the blink detection, the set-up shown in Figure 7 is used. The ASIC is still powered using the NFC tag. The mock-up eye is placed at angles *θ* ≈ *φ* ≈ 0°. The servomotor is programmed to apply a series of blinks with various durations. Two blinks 300 ms apart with duration of about 270 ms with closing and opening times of 45 ms and 100 ms, respectively. These values correspond to unconscientious blinks. Figure 10 shows the *θ* angles the eye-tracker provides in time, top graph. Before the eyelid first closing, the correct *θ* value is obtained. When the eyelid closes, the *θ* value changes rapidly, with the lowest value when the artificial eyelid completely covers the IR light source. Since the four PTDs receive the same amount of attenuated IR light, the centroid returns to its initial value as expected. The process repeats itself for the second blink. The eyewear applies the *z*-score algorithm on the received data to detect the occurrence of openings and closings. Plotting the detection based on the *z*-score in time yields the series of pulses in the lower graph of Figure 10. Therefore, blinks are detected and their duration can easily be computed to determine if they correspond to intentional or unintentional ones. As a simple example, a blink duration longer than 0.5 s, interpreted hence as an intentional one, is used to validate the target designation.

Partial eyelid closures are detected as a single glitch, equivalent to the start pulse, and can be thus either discarded or taken into account.

## 4. Discussion

The results presented show that the proposed cameraless eye-tracker is accurate and robust. Several ASICs have been tested; they all have the same performance. The total power consumption of the electronics required by the instrumented SCL is 417 µW, less than half the power harvested (920 µW), out of which 280 µW consumed by the NXP tag. Encapsulating two integrated circuits into the SCL was not convenient at this stage. Hence, further work will consist in integrating into the ASIC only the necessary parts of the tag to harvest energy and to communicate with the NFC reader. There is plenty of available surface area on fabricated ASIC as 61% of chip core is available (see Figure 3). This should further reduce the total power consumption of the on-lens system. Table 1 summarizes the system main features and performances.

Table 2 compares the eye-tracking performance to that of other mobile eye-trackers. The accuracy is more than doubled with a precision about two orders of magnitude better. Moreover, accuracy and precision remain valid after one hour of functioning. Although tested under different conditions, the results are encouraging and show a real potential for the proposed solution.

Moreover, we demonstrated that the eyelid only briefly disturbs the computed gaze direction when closing or opening and therefore can be used as a command to top the gaze detection recording or validate a target provided that conscious and unconscious blinks can be differentiated. These short disturbances are used to implement simply a bio-monitoring of the instrumented contact lens by interpreting the blinks, provided to keep aside reflex or unconscious blinks. As explained, this can be based on the blink duration; any with duration shorter than 0.5 s can be discarded. Based on duration, one can imagine coding several commands using series of voluntary blinks, one blink corresponding to command 1, two blinks to command 2, and so forth. However, using different voluntary blink durations is not convenient to implement several commands, even though discriminating different duration is technically not a problem. Indeed, it is difficult to ensure that the user blink with the correct duration (even considering a possible discrepancy in the duration). This could yield wrong interpretation and hence the wrong command activated.

A better solution is to use both eyes, each equipped with an instrumented SCL. As unconscious blinking occurs on both eyes at the same time, one blink on one eye is necessarily a command. This increases the number of possible combinations if coupled to series of blinks (left, right, both eyes). An example of three blink-controlled commands is given in Table 3.

The *z*-score algorithm has the advantage of being simple to implement and yields satisfactory results with the mock-up eye. The right threshold value was determined from a series of experiments; it might not be adapted when the system will be used on a real eye. Moreover, a fixed threshold is not appropriate considering almost certain user-to-user and devices variations. The threshold should be tailored by means of a specific algorithm. The same approach also works for a partial blink, although if the eyelid only covers the two upper photodiodes (fully or partially), a unique peak will be detected. For this reason and for sake of simplicity, the HMI should rather be based on the differential response between the two eyes as proposed.

As the system is intended to be worn, ensuring eye safety is mandatory. Bio-compatibility is guaranteed by encapsulating all components (flexible substrate, ASIC, PTDs, etc.) in a medical SCL. Moreover, this rigid lens rests on the sclera and thus creates a tear-filled vault between the SCL and the cornea, improving comfort and avoiding eye dryness. The eyewear continuously illuminating the eye, an IR source is chosen to avoid disturbing the user. The measured irradiance at the lens surface is less than 20 µW/mm^2^, compliant with ocular safety standards for near-IR lighting [15]. Regarding the induction link, the specific absorption rate (SAR) must be less than 2 W/kg. Hence, for safety reason it is necessary to limit the primary-antenna-emitted power imbedded in our case into the eyewear frame, thus at a fixed distance of 1.5 to 2 cm from the eyes. The only variable parameter is the orientation changing when the eye rotates. The rotation of the secondary coil with respect to the primary one, from 0° to 16°, leads to a 30% loss but with no impact on the system harvesting. Simulations in a similar case (but with no rotation) show a SAR of 0.021 W/kg with a 2 W power source [16]. Therefore, we can conclude the 140 mW power source used in this study complies with safety standards. Moreover, studies show prototypes of electronic lens encapsulating a LED, consuming up to 10 mW and worn by humans without a significant rise in temperature. The system presented, consuming less than 0.5 mW, should only moderately generate heat and be compliant with human safety.

Finally, the current system could also be used to detect fatigue since blink duration, interval, and partial closure is a symptom of fatigue or drowsiness [17,18]. This requires just modifying the software program on the eyewear to obtain these data from the start and end pulses detected by the system.

## Figures and Tables

**Figure 1 sensors-22-00595-f001:**
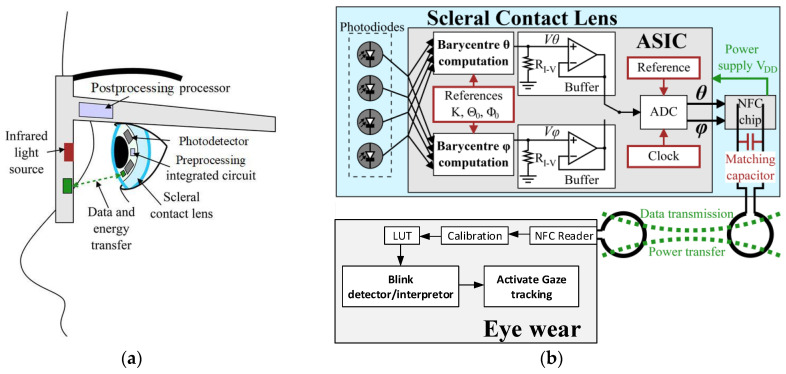
(**a**) Multifunction-instrumented SCL and its eyewear. An inductive link is used to transmit data and powers the SCL. (**b**) Block diagram of both elements; the eyewear performs data postprocessing to use blink as a HMI.

**Figure 2 sensors-22-00595-f002:**
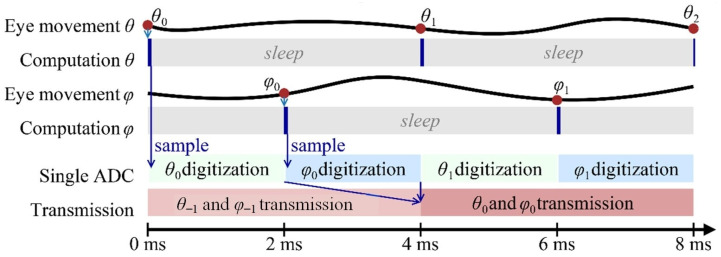
Time diagram for our system. Centroid computing units compute *θ* and *φ* are resting most of the time to save power. The results are sequentially sampled at 500 Hz.

**Figure 3 sensors-22-00595-f003:**
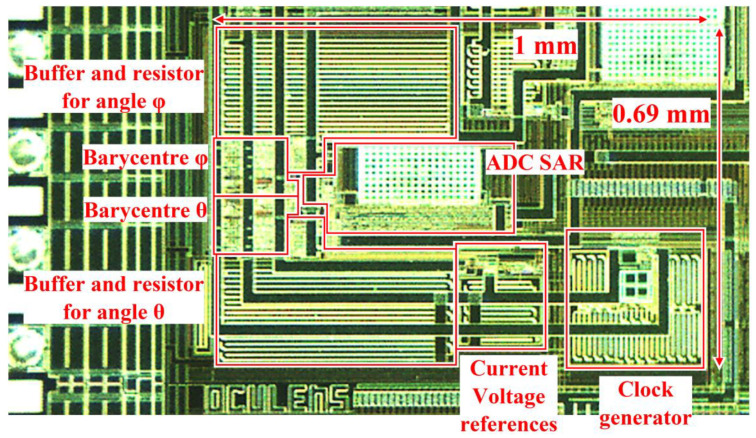
Microphotograph of the fabricated 0.35 µm CMOS ASIC. The circuit supply voltage is 1.8 V and has an active surface area of 0.69 mm^2^.

**Figure 4 sensors-22-00595-f004:**
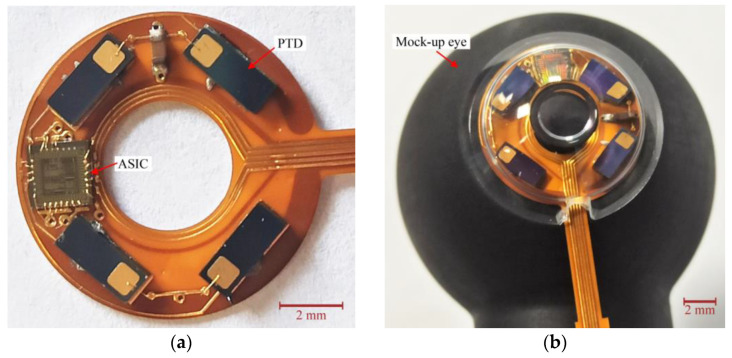
Prototype of the SCL electronics: (**a**) before encapsulation, (**b**) encapsulated.

**Figure 5 sensors-22-00595-f005:**
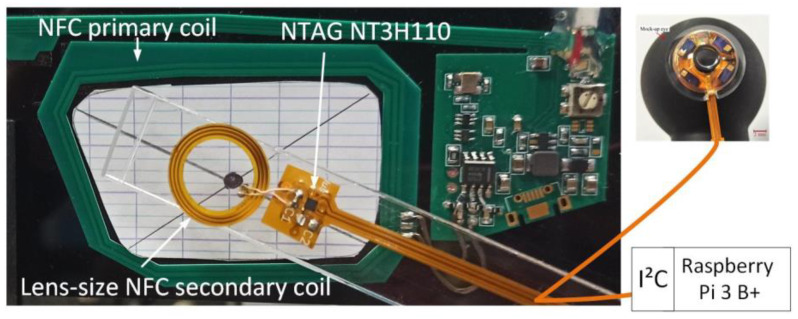
Powering the scleral lens by mean of NFC. The flexible ribbon connects the NFC chip to the encapsulated electronics to ease prototyping and the Raspberry card to retrieve the gaze direction.

**Figure 6 sensors-22-00595-f006:**
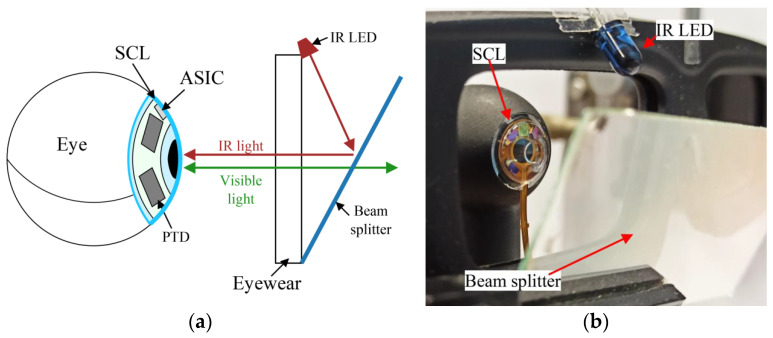
Deporting IR light source on the eyewear: (**a**) principle, (**b**) actual realization.

**Figure 7 sensors-22-00595-f007:**
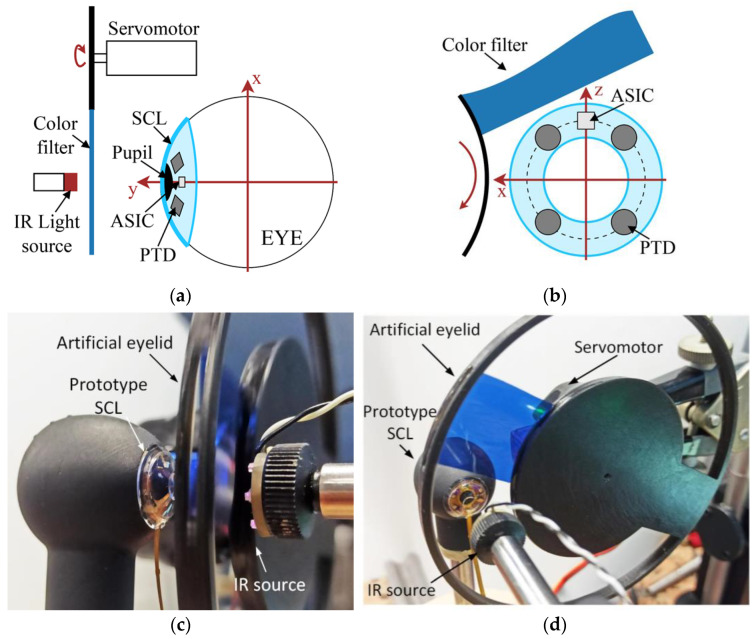
Eyelid emulator principle: (**a**) side view, (**b**) front view. Actual implementation: (**c**) side view, (**d**) front view.

**Figure 8 sensors-22-00595-f008:**
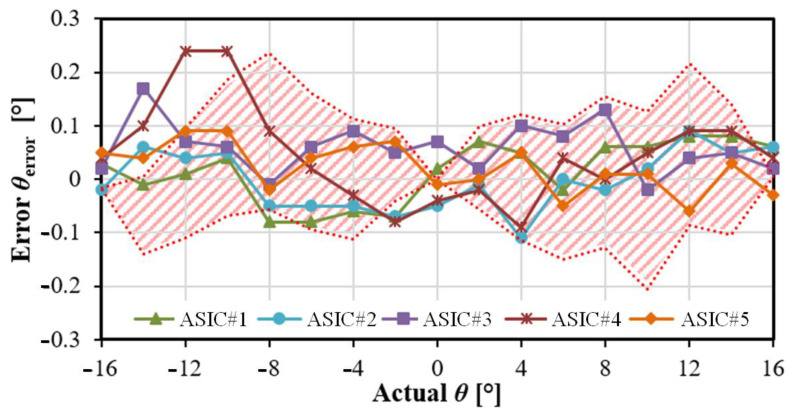
Measured accuracies from 5 ASICs, compared to MC analysis (hatched area); the supply is nominal, i.e., 1.8 V.

**Figure 9 sensors-22-00595-f009:**
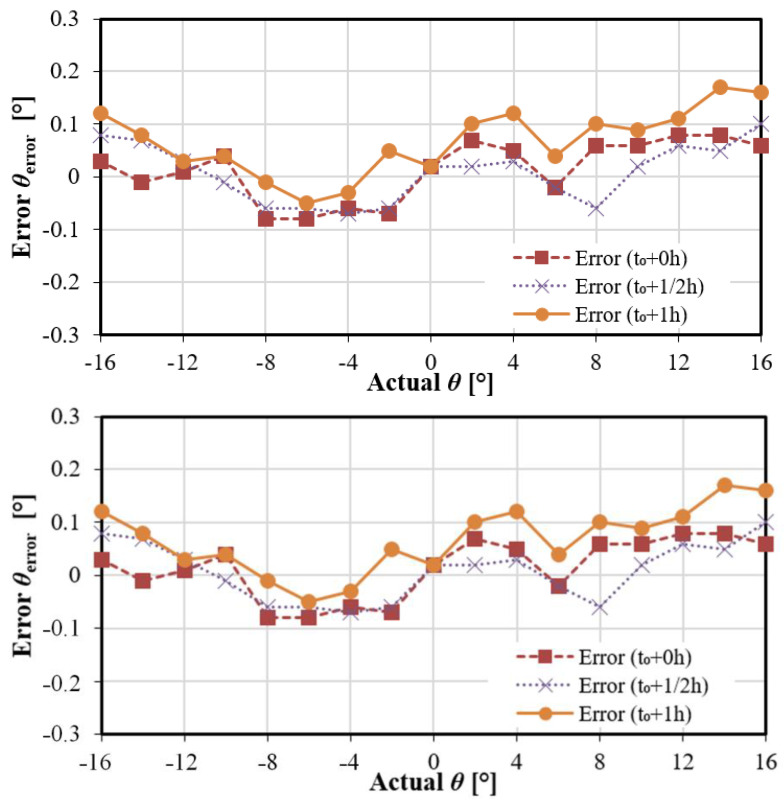
Calibration robustness versus operating time. Calibration done once at t_0_. Accuracy remains stable even after 1 h without recalibration. The supply is nominal, i.e., 1.8 V.

**Figure 10 sensors-22-00595-f010:**
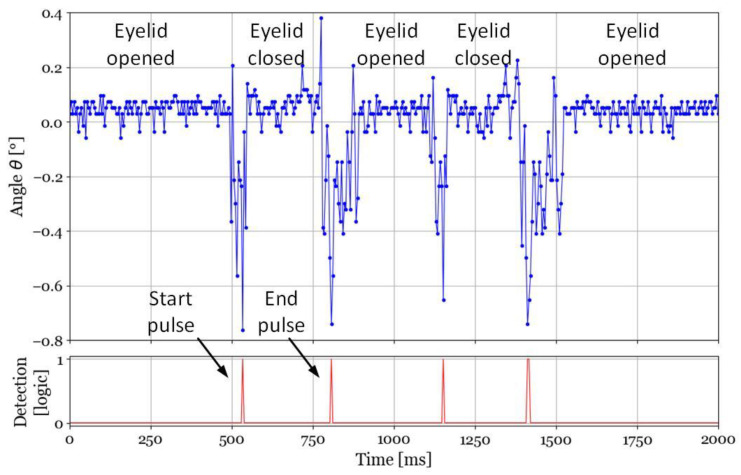
Computed gaze direction versus time. Blink effects on the computation are used to generate blink start and end pulses to establish the blink duration so that the system interprets it as either a reflex or an intentional one.

**Table 1 sensors-22-00595-t001:** Eye-tracker main features and performances.

Item	Value	Unit
ASIC power consumption	137	µW
ASIC active surface area	0.69	mm^2^
ASIC supply voltage	1.8	V
NTAG power consumption	280	µW
Harvested power	920	µW
Sample rate	250	Hz
Eye-tracker accuracy	0.2	degrees
Eye-tracker precision	0.04	degrees

**Table 2 sensors-22-00595-t002:** Eye-tracker main features and performances.

	This Work	Tobii Pro Glasses 2 [6]	Pupils Labs Glasses [7]
Test conditions	On test bench	Worn by user in controlled environment	Worn by user in controlled environment
Accuracy	0.2°	0.5°	0.82°
Precision	0.04°	0.3°	0.31°
Sample rate	250 Hz	100 Hz	240 Hz
Accuracy loss after calibration	None 1 h after	Unknown	+0.25° 5 min after

**Table 3 sensors-22-00595-t003:** Combining different voluntary blinks to generate three commands: “1” means one blink, “0” no blink.

Right Eye	Left Eye	Command Number
1	0	1
0	1	2
1	1	3

## Data Availability

Not applicable.

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
