# Peer review of "Multipurpose Bio-Monitored Integrated Circuit in a Contact Lens Eye-Tracker"

_sensors, 2022, doi:10.3390/s22020595_

Round 1

Reviewer 1 Report

  • This manuscript is extend research from Ref.[9], the authors are encouraged to summarize difference and progress compared with Ref. [9].
  • This submission summarize the system performance in Table I, it would be necessary to compared performance of present techniques. The authors can also use comparisons listed in Ref.[9].
  • For safety regulation, the authors use much information listed in Ref.[17] as reference to illustrate safety of this paper. But, the electrical and magnetic field distributions are highly environmentally dependent when system is operated in near field. It is definitely necessary to setup simulation model of proposed system to investigate EM behaviors instead of using reference data listed in Ref.[17].

Author Response

This manuscript is extend research from Ref.[9], the authors are encouraged to summarize difference and progress compared with Ref. [9].

We have modified the last paragraph of the introduction to better reflect the progress made in this work compared to [9]

This paper extends works presented in [9] by adding this HMI to the system. The HMI is implemented using the ASIC designed to continuously computed the gaze direction. The blinking command is detected on the eyewear from the data sent by the SCL by a simple algorithm. By analyzing blink durations or blink sequences on both eyes, a voluntary command can be detected. This command could be used to validate whether the gaze direction data are recorded or not or to validate a target designation. The number of possible combinations is increased when considering both eyes to design more complex HMIs. Moreover, we present performance measurements (accuracy, precision, robustness, energy harvesting, …) of the full system (ASIC encapsulated into a scleral lens and eyewear) in realistic test setups whereas [9] presented only simulation results of the ASIC.

This submission summarize the system performance in Table I, it would be necessary to compared performance of present techniques. The authors can also use comparisons listed in Ref.[9].

  • Our aim was not comparing our device with conventional eye-trackers, even if it can be used for. Detecting the direction of sight offers various functionalities different from eye-tracker ones (such as pupil size measuring, PERFOS etc.). Our device is rather an eye pointer or cursor. Compared to image processing tasks performed by conventional eye-trackers, its detection process is very fast (no image processing), accurate (0.2° among the best performance) and simple. This difference has been highlighted in the introduction.

We have added the table below in the Discussion section (line) and a small paragraph. 2 references have also added to the paper.

Table 2. Eye-tracker main features and performances.

This work

Tobii Pro Glasses 2

[16]

Pupils Labs glasses

[17]

Test conditions

On test bench

Worn by user in controlled environment

Worn by user in controlled environment

Accuracy

0.2°

0.5°

0.82°

Precision

0.04°

0.3°

0.31°

Sample rate

250 Hz

100 Hz

240 Hz

Accuracy loss after calibration

None 1h after

Unknown

+ 0.25° 5 min after

Table 2 compares the eyetracking performance to that of other mobile eyetrackers. The accuracy is more than twice better with a precision about two orders of magnitude better. Moreover, accuracy and precision remain valid after one hour of functioning. Although tested under different conditions, the results are encouraging and show a real potential for the proposed solution

References added:

  1. Cognolato ,M.; Atzori ,M.; and Muller, H.; “Head-mounted eye gaze tracking devices : An overview of modern devices and recent advances”, J. Rehabil. Assistive Technol. Eng. 5 June 2018), p. 1-13.
  2. Ehinger, B. V.; Groß, K.; and Ibs, I.; “A new comprehensive eye-tracking test battery concurrently evaluating the Pupil Labs glasses and the EyeLink 1000”, PeerJ 7 :e7086 (Apr. 2019), p. 1-43.

  • For safety regulation, the authors use much information listed in Ref.[17] as reference to illustrate safety of this paper. But, the electrical and magnetic field distributions are highly environmentally dependent when system is operated in near field. It is definitely necessary to setup simulation model of proposed system to investigate EM behaviors instead of using reference data listed in Ref.[17].

Energy harvesting efficiency by electromagnetic induction depends upon few parameters such as coil shape and geometry matching, distance and orientation. For safety reason it is necessary to limit the primary antenna emitted power. This one is here embedded in the eyewear frame, hence the distance is fixed between 1.5 to 2 cm far from the eyes. The only variable parameter is the orientation changing when the eye rotates. The rotation of the secondary coil (w. r. to the primary antenna) from 0° to 16° leads to a 30% loss but with no impact on the system harvesting. The energy transfer power emitted by the primary antenna is in our case 340mW, in agreement with the maximum SAR for a person (see [19], previously [17]). This point has been clarified in the paper as we have modified the end the Discussion section as followed:

Hence, for safety reason it is necessary to limit the primary antenna emitted power imbedded in our case into the eyewear frame, thus at a fixed distance of 1.5 to 2 cm from the eyes. The only variable parameter is the orientation changing when the eye rotates. The rotation of the secondary coil, with respect to the primary one, from 0° to 16° leads to a 30% loss but with no impact on the system harvesting. Simulations in a similar case (without rotation) show a SAR of 0.021 W/kg with a 2 W power source [19]. Therefore, the 140 mW power source used in this study complies with safety standards.

Reviewer 2 Report

none

Author Response

N/A

Reviewer 3 Report

  1. In page 6, line 185, the title need to correct.
  2. In page 7, line 207, Figure 6 (c) may be wrong need to correct.
  3. In page 11, line 341 to line 345, need to rearrange.

Author Response

We have modified the 3 typos you noticed